# The Major Factors Causing the Microspore Abortion of Genic Male Sterile Mutant *NWMS1* in Wheat (*Triticum aestivum* L.)

**DOI:** 10.3390/ijms20246252

**Published:** 2019-12-11

**Authors:** Junchang Li, Jing Zhang, Huijuan Li, Hao Niu, Qiaoqiao Xu, Zhixin Jiao, Junhang An, Yumei Jiang, Qiaoyun Li, Jishan Niu

**Affiliations:** 1National Centre of Engineering and Technological Research for Wheat/Key Laboratory of Physiological Ecology and Genetic Improvement of Food Crops in Henan Province, Henan Agricultural University, Zhengzhou 450046, China; jcli@stu.henau.edu.cn (J.L.); jzhang1023@126.com (J.Z.); lhj19960901@163.com (H.L.); 15838232762@163.com (Q.X.); zxjiao2018@163.com (Z.J.); jhan68@163.com (J.A.); nxyjym@henau.edu.cn (Y.J.); liqiaoyun@henau.edu.cn (Q.L.); 2Institute of Cotton Research, Chinese Academy of Agricultural Sciences, Anyang 455000, China; m15927306510@163.com

**Keywords:** wheat (*Triticum aestivum* L.), genic male sterility, microspore, development, molecular regulation

## Abstract

Male sterility is a valuable trait for genetic research and production application of wheat (*Triticum aestivum* L.). *NWMS1*, a novel typical genic male sterility mutant, was obtained from Shengnong 1, mutagenized with ethyl methane sulfonate (EMS). Microstructure and ultrastructure observations of the anthers and microspores indicated that the pollen abortion of *NWMS1* started at the early uninucleate microspore stage. Pollen grain collapse, plasmolysis, and absent starch grains were the three typical characteristics of the abnormal microspores. The anther transcriptomes of *NWMS1* and its wild type Shengnong 1 were compared at the early anther development stage, pollen mother cell meiotic stage, and binucleate microspore stage. Several biological pathways clearly involved in abnormal anther development were identified, including protein processing in endoplasmic reticulum, starch and sucrose metabolism, lipid metabolism, and plant hormone signal transduction. There were 20 key genes involved in the abnormal anther development, screened out by weighted gene co-expression network analysis (WGCNA), including *SKP1B*, *BIP5*, *KCS11*, *ADH3*, *BGLU6,* and *TIFY10B*. The results indicated that the defect in starch and sucrose metabolism was the most important factor causing male sterility in *NWMS1*. Based on the experimental data, a primary molecular regulation model of abnormal anther and pollen developments in mutant *NWMS1* was established. These results laid a solid foundation for further research on the molecular mechanism of wheat male sterility.

## 1. Introduction

Wheat (*Triticum aestivum* L.) is one of the most important food crops in the world. With rapid growth of the global population and reduction in cultivated land area, increasing wheat production has become the need of the hour. Wheat productivity will arguably have more influence on global food security than any other crop [1,2,3,4]. Male (anther) development is pivotal for pollen fertility and crop production. Improving the understanding of pollen development may help increase grain yields and facilitate crop breeding [5]. Abnormal anther development may result in male sterility. Stable hereditary male sterility mutants are valuable germplasm resources for heterosis utilization, which is one of the key ways to improve wheat yield [6]. Male sterility permits the production of hybrids on a commercial scale, relying on heterosis in crops, and can greatly increase selection efficiency of the yield [7]. In-depth understanding of the molecular mechanism of male sterility can help us to use it more effectively.

Plant male sterility was first reported in the 18th century [8]. A large number of plant male sterility mutants have been reported over the past few decades. Male sterility includes cytoplasmic male sterility (CMS) controlled by mitochondrial genes coupled with nuclear genes, and genic male sterility (GMS) controlled by nuclear genes alone [9,10,11]. The pistils of the cytoplasmic male sterile lines and genic male sterile lines are normal, but their stamens are abnormal and cannot pollinate normally, which is due to stamen degeneration, pollen abortion, or functional sterility. GMS occurs in 216 species and 17 species crosses, while CMS occurs in more than 150 species and 271 species crosses [11,12,13]. Plant male sterility lines have been widely studied and applied successfully in many crops, such as maize (*Zea mays*), rice (*Oryza sativa*), soybean (*Glycine max*), and barley (*Hordeum vulgare*). Several wheat CMS lines have been studied and applied in production; however, commercial application is limited [14,15,16]. Possible applications of GMS in plant breeding have been reviewed and discussed for at least 30 years [17]. Up to now, only five GMS genes have been reported in bread wheat. They are *ms1* on 4BS [18], *Ms2* on 4DS [19], *Ms3* on 5AS [20,21], *Ms4* on 4DS [22], and *ms5* on 3AL [23]. Among them, *ms1* and *ms5* are recessive genes, and *Ms2*, *Ms3* and *Ms4* are dominant genes. There are seven alleles on *Ms1* locus: *ms1a*, *ms1b*, *ms1c*, *ms1d*, *ms1e*, *ms1f*, and *ms1g* [23,24]. Three genes—*Ms1*, *Ms2*, and *Ms5—*have been cloned; *Ms1* and *Ms5* encode a glycosylphosphatidylinositol (GPI)-anchored lipid transfer protein and *Ms2* encodes an orphan protein [25,26,27,28]. The Ms1 is required for microgametogenesis in wheat. The orthologs of *ms1* in the A and D subgenomes of wheat are epigenetically silenced, only *Ms1* from the B subgenome expresses [26]. Knockout of *Ms1* generates male-sterile lines [29]. Wheat *ms5* male sterility is induced by recessive homoeologous A and D genome non-specific lipid transfer proteins [28]. Only the *Ms2* mutant is currently widely used in wheat breeding [27]. Identifying more wheat male sterility genes and understanding their functions are the highest priorities for hybrid breeding.

Although the molecular mechanisms of anther and pollen development in model species, such as Arabidopsis and rice, are well understood, little is known about the equivalent processes in wheat [16]. Wheat sporophytic tapetum cell specifically expressed genes *TAA1* (*T. aestivum* anther) and *RAFTIN* (structural protein) are essential for pollen development [30,31]. *MS26/CYP704B* is required for anther and pollen wall development in many species, such as maize, barley, rice, and sorghum (*Sorghum bicolor*) [32,33]. The maize *Male sterile 45* (*Ms45*) gene encodes a strictosidine synthase-like enzyme, which is required for male fertility. The homologs of *MS26/CYP704B* and *Ms45* in wheat have similar functions, and mutations in the A, B and D homeologs lead to male sterility [34,35]. Cold stress contributes to aberrant cytokinesis during pollen mother cell meiosis I in wheat thermosensitive GMS line BS366. The miR167 and tasiRNA-ARF (trans-acting small interfering RNAs-Auxin-Responsive Factor) play roles in regulating the auxin-signaling pathways, which are linked with male sterility in the temperature GMS line BS366 during cold stress [36,37]. Wheat AGL6 transcription factors interact with *TaAP3*, *TaAGAMOUS*, and *TaMADS13*. *TaAGL6* plays an essential role in stamen development through transcriptional regulation of *TaAP3* and other related genes [38]. These researches have enriched the mechanism study of wheat male sterility.

Although plant male development mechanisms are similar, because of the diverse floral phenotypes of various species, the molecular mechanism of wheat male fertility is unique and complicated, and there are many issues that need to be elucidated. Previously, we obtained a stable hereditary male sterility mutant, named *NWMS1*, from the EMS-treated wheat cultivar Shengnong 1 [39]. This study reports molecular mechanisms for the microspore abortion of the male sterility line *NWMS1.*

## 2. Results

### 2.1. Morphology of the Mutant NWMS1 and Shengnong 1

The mutant *NWMS1* (Figure 1a) was mutagenized from wheat cultivar Shengnong 1 (Figure 1a). Most agronomic traits of *NWMS1* were similar to that of Shengnong 1, including plant height, spikelet number on the main stem, and effective tiller number, among others (Figure 1a and Table 1). However, the developmental stages of *NWMS1* were delayed by about 7 days. In the field, at the anthesis stage, when the florets of *NWMS1* were beginning to open, the anthers of Shengnong 1 were exposed and the glumes were closed well (Figure 1b). The glume open angle of *NWMS1* was larger (Figure 1d). The leaf apex of *NWMS1* was abnormal from the seedling stage—it was elliptical and light yellow in colour in the heading stage (Figure 1c). The leaf apex trait can be used as a marker of male sterile plants at early stages.

The spikes of *NWMS1* and Shengnong 1 were continuously observed with appearance of the spike primordia. There were no significant differences before Zadoks Stage 32 (Vahamidis Stages 12). The spikelet primordia began to differentiate, and distinct pistils and anthers were visible from Zadoks Stage 32 (Vahamidis Stages 12). The anthers of *NWMS1* were smaller than that of Shengnong 1. The normal anthers of Shengnong 1 gradually matured and released their pollens. The abnormal anthers of *NWMS1* were complete without dehiscence and no pollen grains were released to the pistils (Figure 1h–k,o–r). The anthers of *NWMS1* dried in the end (Figure 1r).

### 2.2. Anther and Microspore Structures of Mutant NWMS1 and Shengnong 1

The pollen mother cells (PMCs) of *NWMS1* and Shengnong 1 were similar (Figure 2A–E,a–e). The meiosis appeared to be normal in *NWMS1*. Their PMCs divided normally to form microspore mother cells, underwent meiosis, and formed dyads, tetrads, and tri-nucleate microspores (Figure 2C,c). The two fusiform sperm nucleis and one round vegetative nucleus in the early trinucleate stage were found to be normal (Figure 2D,d). Germinal apertures of the microspores appeared normal. However, the sperm nucleis of the sterile microspores were gradually degraded and disappeared (Figure 2E,e).

A number of pollen grains were observed in the anthers of *NWMS1*, which was distinguished from Taigu genic male-sterile wheat (*Ms2*; [27]). The pollen viability was investigated by staining with 0.5% 2,3,5-triphenyltetrazolium chlorid (TTC) and 1% I_2_-KI. The difference between Shengnong 1 and *NWMS1* was clear when stained with TTC (Figure 2F,f). Most pollen grains of Shengnong 1 were strongly strained, but pollen grains of *NWMS1* were lightly strained with TTC, indicating they had no or very weak viability. Similar results were also reached when the pollen grains were stained with I_2_-KI (Figure 2G,g). The data indicated that although a number of pollen grains existed in the anthers of *NWMS1*, they had no viability. Thus, *NWMS1* was male sterility.

The anthers at Zadoks Stage 59 (Figure 1j,q) were dissected and observed with a scanning electron microscope (Figure 2H–I,h–i) and a transmission electron microscope (Figure 2J–K,j–k). The outer epidermal cells of the fertile anthers were found to be smooth with clear strip lines (Figure 2H,I), and the outer epidermal cells of the sterile anthers were rugged, collapsed with unsharp strip lines (Figure 2h,i). The tapeta were observed clearly in both fertile and sterile anthers under a transmission electron microscope. In the meanwhile, the tapetum layers in fertile anthers gradually degraded during pollen maturating. Besides, the middle layer cells in fertile anthers disappeared, and the middle layer cells in sterile anthers still existed at the pollen mature stage (Figure 2J,j). However, the pollen grains of Shengnong 1 and *NWMS1* were significantly different. The fertile pollen grains of Shengnong 1 were plump and spherical with many starch grains, which indicated that there was rich nutrient accumulation (Figure 2K). The pollen grains of *NWMS1* were found to be ellipse; severe plasmolysis occurred, and they had no starch grains. Some autophagosomes appeared at the edge of the large vacuoles (Figure 2k).

These results demonstrate that the microspore abortion of *NWMS1* began to occur from the binucleate stage. The male sterility was caused by the degradation of the microspore nuclei and other functional substances, as well as no starch grain accumulation, which distinguished between the reported genic male sterile mutants in wheat. This result suggests that *NWMS1* was a new genic male-sterile mutant.

### 2.3. PCD Detection in Anthers

In order to discover whether programmed cell death (PCD) was one of the reasons for pollen abortion in *NWMS1*, we analyzed the cleavage of nuclear DNA using the TUNEL assay in Shengnong 1 and mutant *NWMS1* when the microspores were at the binucleate stage (Figure 3). The nucleus stained with DAPI was red. Most pollen grains of Shengnong 1 were rounded (Figure 3A). Most pollen grains of mutant *NWMS1* were deformed and their cell nuclei had degraded (Figure 3a). The apoptotic cell nuclei were stained deep green with the TUNEL kit. It was clear that apoptotic cells mainly existed in anther wall layers, and there was no significant difference between Shengnong 1 and mutant *NWMS1* (Figure 3B,b). None of the cell nuclei of pollen grains in Shengnong 1 were stained green, pointing towards no cell apoptosis. The cell nuclei of the few almost spheral pollen grains in mutant *NWMS1* were stained deep green, indicating cell apoptosis (Figure 3C–D,c–d). Obviously, most pollen grains of *NWMS1* were degraded, only few pollen grains remained spheral, and cell apoptosis had already taken place; this resulted in complete male sterility.

### 2.4. Overview of the Transcriptome Data

A total of 215.69 Gb clean data were obtained from the 18 libraries (S1_*NWMS1*: T01, T02, T03; S1_WT: T04, T05, T06; S2_*NWMS1*: T07, T08, T09; S2_WT: T10, T11, T12; S3_*NWMS1*: T13, T14, T15; S3_WT: T16, T17, T18), at least 10.24 Gb clean data for each sample; the average Q30 percentage was more than 93.42%. The reads were compared with the *T. aestivum* reference genome, and the mapping ratio varied from 85.70% to 93.55%. Based on the mapped results, 42,963 novel genes were identified, 27,267 of them functionally annotated.

The three-dimensional map of PCA clearly showed a good correlation among the transcriptome sequencing samples (Figure 4a). The colored points in the PCA map represented different samples. The three eigenvectors (69.1%, 16.5%, and 8.3%) clearly separated these samples, which was consistent with the morphological pattern of the different tissues. PCC analysis showed that each correlation coefficient was more than 0.98 between all replicated samples (Figure 4b), which demonstrated that the biological replicates were highly consistent.

A total of 45,048, 44,849, 48,663, 48,590, 54,608, and 46,937 valid genes were identified from the 6 samples, respectively (Figure 5a). The number of the valid genes expressed only in S1_*NWMS1*, S1_WT, S2_*NWMS1*, S2_WT, S3_*NWMS1*, and S3_WT were 824, 756, 1432, 831, 1886, and 4297, respectively (Figure 5b). This result showed that the number of the valid genes was gradually increased in various tissues with spike development (the differentiation from S1 to S3).

### 2.5. DEGs among the Six Samples

The differentially expressed genes (DEGs) were identified by pairwise comparisons of the 18 libraries. When FDR < 0.001 was set as the changeless filtering threshold, the numbers of DEGs identified among the six tissues at the three stages were 32,833, 16,152, 9968, and 6779 at different fold changes (Figure 5c). The number of DEGs decreased with increase in fold change. The number of DEGs in the young spikes at the early anther developmental stage (S1_*NWMS1* and S1_WT) was always the least. The stamens of *NWMS1* appeared normal (Figure 1f,m), and the number of DEGs was small. The valid genes and DEGs were the most in anthers at the binucleate stage (S3_*NWMS1* and S3_WT), which was consistent with the microspore abortion that occurred at this time (Figure 1j,q and Figure 2D–E,d–e).

In order to further explore the key genes involved in abnormal pollen development in *NWMS1*, significant DEGs (Log_2_FC ≥ 3 or ≤ −3) were screened between the *NWMS1* and WT. A total of 1402,170 and 7658 DEGs were identified (Appendix A and Figure 5d) (FDR < 0.001 and FC ≥ 8) in sample pairs of S1-*NWMS1* vs. WT, S2-*NWMS1* vs. WT, and S3-*NWMS1* vs. WT, respectively. These genes were classified into 44 of the 52 subcategories, with reference to the GO database (Appendix A), and classified into six main metabolic pathways, with reference to the KEGG database.

Further enrichment and DEGs analysis according to gene number and *p* value of the main terms in the GO database, top enrichment terms of the DEGs in the biological process, cellular components, and molecular functions are listed in Table 2. They were response to cadmium ion (GO:0046686), cytoplasmic membrane-bounded vesicle (GO:0016023) and electron carrier activity (GO:0009055) for DEGs of S1-*NWMS1* vs. WT; pentose-phosphate shunt (GO:0006098), cytoplasmic vesicle (GO:0016023) and heme binding (GO:0020037) for DEGs of S2-*NWMS1* vs. WT; oxidation-reduction process (GO:0055114), cytoplasmic membrane-bounded vesicle (GO:0016023), and heme binding (GO:0020037) for DEGs of S3-*NWMS1* vs. WT. These enrichment results show that most genes were related to nutrient transport and metabolism.

The top metabolic pathways (Table 3) were classified based on gene annotations in the KEGG database. The DEGs of S1-*NWMS1* vs. WT involving top three enriched pathways were amino sugar and nucleotide sugar metabolism (ko00520). The DEGs of S2-*NWMS1* vs. WT involving top three enriched pathways were phenylalanine metabolism (ko00360), phenylpropanoid biosynthesis (ko00940), and glutathione metabolism (ko00480). The DEGs of S3-*NWMS1* vs. WT involving top three enriched pathways were phenylalanine metabolism (ko00360), glutathione metabolism (ko00480), and starch and sucrose metabolism (ko00500). Some common metabolic pathways were extremely active during wheat anther growth, including phenylalanine metabolism (ko00360) and glutathione metabolism (ko00480).

A total of 33 significant DEGs were identified among the three paired DEGs (Figure 5e). Obviously, most of these genes were less expressed in Shengnong 1, and significantly highly expressed in *NWMS1*, especially at the meiotic stage (S2). Only four DEGs were highly expressed in Shengnong 1. This result implied the expression of these DEGs, especially at the S2 stage, determining the microspore abortion in *NWMS1*.

### 2.6. DEG Co-Expression Clusters

Hierarchical clustering was performed to identify the sample type-specific gene expression profiles of the DEGs from among the 18 libraries (Figure 6a). Nine gene co-expression profile maps were made using the *k*-means cluster, with *k* as 9. A total of 8863 genes were selected and classified into the nine groups (Figure 6b and Appendix A).

Compared with Shengnong 1, clusters K1 and K2 included 3766 genes, which were highly expressed in S2_*NWMS1* and S3_*NWMS1*. The genes in cluster K5 (1661 genes), cluster K6 (1268 genes), and cluster K8 (509 genes) were significantly highly expressed in Shengnong 1 at the binucleate stage (S3 stage), compared to the first two stages. The genes in cluster K3 (715 genes) and K7 (225 genes) were expressed at similar levels in Shengnong 1 and *NWMS1* at S1 and S2 stages. The expression level at S2 was significantly higher than that at S1; their expression level was rapidly lowered at the S3 stage, and the expression level in Shengnong 1 was even lower. From the early anther development stage (S1 stage) to the binucleate stage (S3 stage), the expression levels of the genes in cluster K4 (296 genes) were gradually up-regulated, and that of the genes in cluster K9 (423 genes) were gradually down-regulated.

To further explore the metabolic pathways that these clustered genes participated in, various metabolic pathways were identified, referring to the KEGG database (Appendix A and Figure 6c). In the nine clusters, clusters K1 and K2 had the most genes in the anthers of *NWMS1* at the meiotic stage (S2 stage) and the binucleate stage (S3) stage; their expression patterns were found to be similar. We identified that these genes were clearly involved in phenylalanine metabolism (ko00360), phenylpropanoid biosynthesis (ko00940), and glutathione metabolism (ko00480). Though the genes in clusters K3 and K7 had similar expression patterns in anthers at the meiotic stage (S2 stage) and the binucleate stage (S3), their enriched metabolic pathways were completely different.

A total of nine metabolic pathways were considered as candidate pathways related to male sterility by *K*-means analysis in the end (Figure 6c). They were glutathione metabolism (ko00480) in cluster K1, phenylpropanoid biosynthesis (ko00940) in cluster K2, steroid biosynthesis (ko00100) in cluster K4, starch and sucrose metabolism (ko00500) in cluster K6, pentose and glucuronate interconversions (ko00040) and phosphatidylinositol signaling system (ko04070) in cluster K6, biosynthesis of unsaturated fatty acids (ko01040) in cluster K7, fatty acid metabolism (ko01212) in cluster K7, and pentose and glucuronate interconversions (ko00040) in cluster K8. These metabolic pathways showed a high correlation in corresponding *K*-means clusters.

### 2.7. Valid Gene Co-Expression Network Analysis with WGCNA

The weighted gene co-expression network analysis (WGCNA) of 19236 valid genes (Appendix A) was carried out. Five distinct modules (marked in different colors) were shown in the cluster dendrogram (Figure 7a), in which each tree branch represented a module and each leaf in the branch represented one gene. The module eigengene was the first principal component of a given module and could be considered representative of the module’s gene expression profile. The five modules were highly correlated with the tissue-specific expression profiles of their eigengenes in the six samples (Figure 7b,c). For example, the correlation coefficient between the brown module and S3_WT sample was 0.98, the error was 6e-13. This result showed a high correlation between the module and the sample. Among them, the samples in S1_*NWMS1* and S1_WT were classified together in the blue module, indicating their molecular similarities.

We obtained the top ten KEGG enrichment pathways in the five modules (Figure 7d). The genes in the blue module mainly belonged to pyrimidine metabolism (ko00240), DNA replication (ko03030), purine metabolism (ko00230), and mismatch repair (ko03430). The genes in the brown module mainly belonged to starch and sucrose metabolism (ko00500) and oxidative phosphorylation (ko00190). The genes in the green module mainly belonged to phenylpropanoid biosynthesis (ko00940), plant hormone signal transduction (ko04075), and glutathione metabolism (ko00480). The genes in the black module mainly belonged to photosynthesis (ko00195). The genes in the light cyan module mainly belonged to protein processing in the endoplasmic reticulum (ko04141). This result was similar to that obtained with the *K*-means cluster analysis.

### 2.8. Key Genes Involved in Abnormal Anther Development of NWMS1 Screened via WGCNA

Genes whose expressions were highly correlated with anther development at specific stages of the samples were potential candidates for the male sterility of *NWMS1*. The light cyan, black, green, and brown modules were highly correlated with S2-*NWMS1*, S2-WT, S3-*NWMS1*, and S3-WT, suggesting their important roles in anther development (Figure 7b). A total of 676 DEGs (Appendix A) were identified from 19,236 valid genes by WGCNA. We screened out the top 100 genes from different modules to establish a network by kME, respectively. Each node represented a gene, and the connecting lines (edges) between genes represented gene co-expression correlations. The significant co-expression genes were selected and ranked by node size, and the four key modules were visualized (Figure 8). The top five key genes in each of the four key modules were screened out. These 20 genes were considered as highly related genes involved in abnormal anther development of *NWMS1*, suitable for further analysis (Table 4). The annotation of the genes mainly referred to the Swiss-prot database (https://www.uniprot.org).

Only seven genes were annotated in the KEGG database (Appendix A). Genes *SKP1B* (TraesCS2D01G420100) and *BIP5* (TraesCS7A01G430600) were involved in protein processing in endoplasmic reticulum (ko04141), genes *KCS11* (TraesCS1A01G413400) and *ADH3* (TraesCS1D01G376300) were involved in fatty acid elongation (ko00062) and fatty acid degradation (ko00071); gene *BGLU6* (TraesCS4B01G248600) was involved in starch and sucrose metabolism (ko00500), and gene *TIFY10B* (TraesCS4D01G296000) was involved in plant hormone signal transduction (ko04075). These key genes obtained via WGCNA were also identified in clusters K2, K3, K4, and K6 by *K*-means clustering (Figure 6). Clearly, the expression profiles of the related genes were the major factors determining the abnormal anther development of *NWMS1*. The important genes involved in key enrichment pathways were screened out, including *SKP1B*, *BIP5*, *KCS11*, *ADH3*, *BGLU6*, and *TIFY10B.*

### 2.9. The Expression Profiles of Key Genes Involved in Anther Development

In order to further demonstrate the reliability of the sequencing results, nine randomly selected genes from different K-means clusters and three known abnormal anther development-related genes (CYP703A2, CSA, and QRT3) were taken to perform quantitative RT-PCR (Figure 9). The experimental samples were the same as the samples used for RNA-seq. The results showed that the changes of the twelve genes among the six samples were consistent with the results of RNA-seq. As documented, the high expressions of genes CYP703A2, CSA, and QRT3 at the binucleate microspore stage (S3) of mutant NWMS1 contributed to the male sterility. Similarly, the homologous genes of mannan synthase 11, saccharopine dehydrogenase, cysteine proteinase RD21a, and cytochrome b5 were significantly highly expressed at S3, and the homologous genes of temperature-sensitive omega-3 fatty acid desaturase, nucleoside diphosphate kinase III, mitochondrial phosphate carrier protein 3 were significantly lowly expressed at S3 in NWMS1. These abnormal expression profiles suggested that they contributed to male sterility in NWMS1.

## 3. Discussion

### 3.1. NWMS1 Belongs to Binucleate Microspore Abortion

Several studies have demonstrated that male sterility occurs at different developmental stages, and the causative factors are various. Many male abortions occur at the early stages of meiosis, the tetrad stage, and the microspore developmental stage in monocotyledonous plants [40]. Pollen abortion can be classified into four types: pollen-free, uninucleate abortive, binucleate abortive, and trinucleate abortive [41]. Wheat Taigu genic male-sterile mutant belongs to pollen-free abortion, which is controlled by *Ms2* [42]. When stained with TTC and KI-I_2_, a number of pollen grains in anthers of *NWMS1* were visible, but they did not have vitality (Figure 2f,g). In many cases, abnormal meiosis and premature tapetum degradation are direct factors causing pollen abortion in male sterility lines [43,44]. The pollen mother cells, meiosis, tapetum, and binucleate microspores of *NWMS1* appeared normal (Figure 2a–d,j); later, the microspores collapsed and severe plasmolysis occurred (Figure 2e,j,k). The anther outer epidermis of *NWMS1* was significantly different from that of Shengnong 1. The nuclei and cytoplasm of the microspores in *NWMS1* were degraded and apoptosis occurred at the binucleate stage (Figure 3). This data suggests that *NWMS1* belonged to binucleate microspore abortion. How they affect pollen grain development needs further research.

### 3.2. Global Gene Transcription Changes in NWMS1 and Shengnong 1

Many factors can cause abnormal male development in plants. Male development includes a number of events, including stamen meristem specification, generation of sporogenous cells, and differentiation into microspore mother cells (MMCs), meiosis, microspore maturation, and pollination [45]. Errors in any of the events results in male sterility. Here, we screened out a series of genes related to male sterility (Table 5).

Meiosis is a crucial event for sexual reproduction of eukaryotes to form haploid spores and gametes. *PAIR1* (*homologous pairing aberration in rice meiosis1*) is required for homologous chromosome pairing and cytokinesis in male and female meiocytes of rice, and PAIR1 protein plays an essential role in establishment of homologous chromosome pairing in rice meiosis [46]. Similarly, the function of PAIR2 is to establish homologous chromosome pairing during rice meiosis [47,48]. *Sgo1* (*shugoshin*) and *Bub1* (for budding uninhibited by *benzimidazole 1*) play important roles in mitosis and meiosis by regulating the spindle formation and chromosome segregation [49,50]. *PSS1* (*pollen semi-sterility1*) encodes a kinesin-1–like protein, which plays an important role in male meiosis, anther dehiscence, and fertility in rice. Cytochrome P450s (P450s) are hemethiolate monooxygenases involved in a vast array of biosynthetic pathways in secondary and primary metabolisms [51,52]. The expression of AtP450, *CYP703A2*, in Arabidopsis is initiated at the tetrad stage and restricted to microspores and to the tapetum cell layer [53]. Arabidopsis *AtMYB80* (formerly *AtMYB103*) is a regulator of tapetal and pollen development; its homologs are functionally conserved in crops [54]. Functional disruption of *AtMYB80* results in male sterility [55,56,57]. *TDR* (Tapetum Degeneration Retardation) is a key component of the molecular network regulating rice tapetum development and degeneration [58]. Arabidopsis mutant *dad1* (*defective in anther dehiscence1*) is defective in anther dehiscence and pollen maturation [59]. Rice *OsGAMYB* is highly expressed in grain aleurone cells, inflorescence shoot apical region, stamen primordia, and tapetum cells of the anther. It plays an important role in floral organ and pollen development [60]. Gibberellins (GAs) regulate exine formation and the programmed cell death (PCD) of tapetal cells via MYB transcription factor GAMYB. Two GA-regulated lipid metabolic genes, a *cytochrome P450 hydroxylase* CYP703A3 and *β-ketoacyl reductase*, may be involved in providing a substrate for exine and Ubisch body [61]. *BAM1* (barely any meristem) and *BAM2* revealed a cell-cell communication process important for early anther development, including aspects of cell division and differentiation [62]. *CSA* (carbon starved anther) encodes an R2R3 MYB transcription factor that is expressed preferentially in anther tapetal cells and in the sugar-transporting vascular tissues. *CSA* can lead to male sterility by reduced levels of carbohydrates in later anthers [63]. *QRT3* (*quartet 3*) is specifically and transiently expressed in the tapetum during the phase when microspores separate from their meiotic siblings, and it plays a direct role in degrading the pollen mother cell wall during microspore development [64]. *OsTDL1A* and *OsTDL1B* are co-expressed with *MSP1* (*multiple sporocytes1*) in anthers during meiosis, and OsTDL1A binds MSP1 in order to limit sporocyte numbers [65]. *OsDEX1* (defective in exine pattern formation) is required for tapetum function and pollen wall formation in rice; OsDEX1 plays a fundamental role in the development of tapetal cells and pollen wall formation, possibly via modulating Ca^2+^ homeostasis during pollen development [66].

Expressions of most homologs of the above described genes in *NWMS1* and Shengnong1 at the three stages (S1, S2 and S3) showed no significant difference. Especially, the genes involved in abnormal anther development showed no significant difference at the early anther development stage (S1) and the meiotic stage (S2). This result is consistent with meiosis observations, which indicate that male sterility is not associated with meiosis. However, the homologs of *CYP703A2*, *CSA*, and *QRT3* were highly expressed in mutant *NWMS1* at the binucleate stage (S3), and the three genes were involved in abnormal microspore differentiation [53,63,64]. Ultramicroscopic observation revealed that the tapetum structure of *NWMS1* was still complete at the pollen grain mature stage (Figure 2j). These results demonstrate that the pollen abortion of *NWMS1* occurred after meiosis and microspore formation.

### 3.3. Complex Network Regulating Anther Development of NWMS1

Wheat anther development is a complex process. Many important metabolism pathways and regulatory molecules are involved in this process, such as sugar and lipid synthesis and transport, phytohormone-signaling, transcription factor, miRNA, and noncoding RNA. Mutation of any of the key genes will lead to male sterility. Some reports have demonstrated that mutations of the genes critical for anther and pollen development will result in male sterility [30,31,34,35,36,37,38,67]. In this study, the metabolic pathways related to male sterility, such as phenylalanine metabolism (ko00360), phenylpropanoid biosynthesis (ko00940), and glutathione metabolism (ko00480), were identified via *K*-means analysis. This result was similar to our previous results of another wheat male sterility line *DMS* [68]. These disturbed pathways were major factors leading to abnormal microspore development in mutant *NWMS1*.

A series of more valuable metabolic pathways and candidate genes related to wheat male development were obtained via WGCNA (Figure 8), including protein processing in endoplasmic reticulum (ko04141), fatty acid elongation (ko00062), fatty acid degradation (ko00071), starch and sucrose metabolism (ko00500), and plant hormone signal transduction (ko04075). Among them, one gene *BGLU6* involved in starch and sucrose metabolism (ko00500), and two genes *SKP1B* and *BIP* involved in protein processing in endoplasmic reticulum were highly expressed in *NWMS1* at the meiotic stage (S2). Disturbances in sugar metabolism and unloading in the anther can significantly influence pollen formation and cause male sterility [69]. Endoplasmic reticulum is the synthesis site of a series of important biological macromolecules in cells, including proteins. Microspore ultrastructure of *NWMS1* showed that the shortage of energy substances was also a direct cause of male sterility at the binucleate stage (S3); this was also reflected at gene expression levels at the meiotic stage (S2). One gene *TIFY10* involved in plant hormone signal transduction (ko04075) was significantly highly expressed in *NWMS1* at the binucleate stage (S3). Researchers have previously demonstrated that both fertility and flower development are controlled in part by jasmonates and fatty acid derived mediators [70,71]. In Arabidopsis, several studies have demonstrated that hindering jasmonate synthesis or occurrence of abnormal jasmonate signal transduction pathway can result in male sterility, such as in *opr3* and *dad1* mutants [59,72]. It has been proposed that jasmonates act by controlling water transport in the anther, and a lack of this hormone in the anther can cause male sterility [59].

This data indicates that sugar and lipid metabolisms, phenylalanine metabolism, phenylpropanoid biosynthesis, and hormone signaling were disturbed in *NWMS1* during microspore development. Functional analysis of these pathways and candidate genes will increase understanding of the male sterility mechanism, and further studies should focus on the link between metabolic pathways and male sterility in wheat.

### 3.4. Starch and Sucrose Metabolism is a Major Factor Causing Male Sterility in NWMS1

Carbohydrates play critical roles in male gametophyte development by providing nutrients for normal growth, and may also influence development as signaling molecules during this process [73,74]. For example, cotton (*Gossypium hirsutum*) GhCK1 is a casein kinase. Overexpression of *GhCK1* exhibits starch synthase kinase activity and influences other factors in cotton flower buds, leading to anther abortion [75]. Rice mutant carbon starved anther (*CSA*) is a male sterile line whose sugar partitioning has been perturbed [63]. In wheat, a decline in invertase activity alters carbohydrate metabolism, resulting in reduced starch accumulation within the pollen, leading to pollen abortion [76]. Cucumber (*Cucumis sativus*) CsSUT1 (sucrose transporter) is a typical plasma membrane-localized energy-dependent high-affinity Suc-H^+^ symporter. The male flowers of *CsSUT1*-RNA interference (RNAi) lines have exhibited a decrease in sucrose, hexose, and starch content, which is highly associated with male sterility [77]. Some genes in Arabidopsis [78], and rice [79,80] are expressed in pollen and support a link between sucrose transports and pollen development and viability.

Starch and sucrose metabolism (ko00500) is the only metabolism pathway screened by both WGCNA and *K*-means classification (Figure 6c and Figure 8a), indicating that the disturbed starch and sucrose metabolism (ko00500) was the most closely related pathway to male sterility of *NWMS1*. This result was confirmed by the phenotype with no starch gain accumulation in pollen grains of *NWMS1*. The pollen grains of mutant *NWMS1* were found to be ellipse, have severe plasmolysis, without starch grain accumulation (Figure 2K,k). Therefore, we speculate that disturbed starch and sucrose metabolism is the most important metabolism pathway causing male sterility in *NWMS1*.

There are many enzymes involved in the wheat starch and sucrose metabolism pathway. Although most enzyme encoding genes were expressed equally in *NWMS1* and WT, some DEGs encoding key enzymes were identified at three stages (Appendix A). The significant DEGs are listed in Table 6. All the DEGs were significantly differentially expressed at S3; only four genes were differentially expressed at S1. Starch is an essential requirement for successful fertilization. Plastidic phosphoglucomutase mutants impair starch synthesis in rice pollen grains, resulting in male sterility [81]. Trehalase is ubiquitous in higher plants. Trehalose and trehalase play key roles in regulating carbohydrate allocation in plants [82]. Sucrose is both energy and a signaling material in plant male development. In summary, disturbed expressions of the genes encoding sucrose synthase, trehalose 6-phosphate synthase, hexokinase, UTP-glucose-1-phosphate uridylyltransferase, alpha-amylase, endoglucanase were considered to play important roles in abnormal male development in *NWMS1*.

## 4. Materials and Methods

### 4.1. Plant Materials and Growth Conditions

Shengnong 1 (WT) is a new wheat line with high yield and powdery mildew resistance, bred and preserved in our lab, the National Centre of Engineering and Technological Research for Wheat, Henan Agricultural University. Mutant *NWMS1* was obtained from Shengnong 1 treated with EMS in 2016 [39].

Shengnong 1 was used as the maintainer of *NWMS1*; the hybrid seeds were planted every year. From wheat growth seasons in 2016, Shengnong 1 and the hybrid seeds of *NWMS1* were planted in our experimental field at Houwang Village, Xingyang City, Henan, P. R. China (34°25′ N, 115°39′ E, 49 m above sea level). The seeds were sown in plots of 3 m in length and 2 m in width. The distance between rows was 0.25 m, and 30 seeds were planted in each row. Fertilizer and weed management was similar to that of wheat breeding [83].

### 4.2. Morphology Observation

The male sterile individuals of *NWMS1* were clear and visible at the booting stage. Whole plants were photographed using a camera (Nikon Coolpix 4500, Nikon Corporation, Tokyo, Japan). The spike primordia, anthers, and pistils of Shengnong 1 and *NWMS1* were observed at different developmental stages, analysed by the Zadoks Scale [84]. The developmental stages of the young spike primordia were described according to the Vahamidis Scale [85]. The florets with anthers and pistils at different developmental stages were dissected from the spikes using an anatomical needle. Then, the anthers of WT and *NWMS1* were observed and imaged using an inverted microscope (SRL-7045A, Beijing Century Science Letter Scientific Instruments Co., Ltd. Beijing, China).

### 4.3. Micrstructure and Ultrastructure Observation

In the early booting stage (Z41) [84], when the distance between the flag leaf ear and the second last leaf ear was less than 4 cm, the spikes were sampled and fixed in Carnoy’s solution (95% ethanol: glacial acetic acid = 3:1). The anthers were separated from the young spikes, mashed with tweezers to release the pollens, and dyed with improved phenol fuchsin solution. The meiosis of the pollen mother cells (PMCs) was observed and imaged with a microscope (OLYMPUSBX-53, Tokyo, Japan). The ultrastructures of the anthers and microscopes were observed using a scanning electron microscope (HITACHI-SU8100, Tokyo, Japan) and a transmission electron microscope (HITACHI-HT7700, Tokyo, Japan).

### 4.4. TUNEL Assay

The anthers of Shengnong 1 and mutant *NWMS1* were fixed in FAA solution (5 mL of formalin, 5 mL of acetic acid and 90 mL of 70% ethyl alcohol). The samples were dehydrated, embedded in paraffin, and sectioned with a rotary microtome, as described by Geng et al. (2018) [86]. Then, paraffin sections were prepared as described by Phan et al. (2011) [87]. The terminal deoxynucleotidyl transferase-mediated dUTP-biotin nick end labeling (TUNEL) assay was performed using a kit (Fluorescein In Situ Cell Death Detection Kit; Roche, Basel, Switzerland), according to the manufacturer’s instructions. The excitation wavelength of the TUNEL assay was 465–495 nm, and emission wavelength was 515–555 nm. The TUNEL kit can stain positive apoptotic cell nuclei green. The excitation wavelength of 4′,6-diamidino-2-phenylindole (DAPI) was 510–560 nm, emission wavelength was 590 nm, and the nucleus stained with DAPI was red under excitation of ultraviolet light.

### 4.5. Sample Preparation and RNA Extraction for RNA-seq

The spikes of Shengnong 1 and *NWMS1* (Figure 1f,m) at Zadoks Stage 32 (Vahamidis Stages 12) were sampled for RNA-seq. The florets had already been differentiated—the length of the young spikes was less than 3 cm, the stem tips (the base section of the spikes) were excluded. This stage was described as the early anther development stage (S1). Each sample had at least 30 individual young spikes. When the spikes (Figure 1h,o) were at Zadoks Stage 41 (Vahamidis Stages 19), the anthers were turning green—the length of the spikes was 3–5 cm, the anthers were sampled for RNA-seq; this stage was described as the meiotic stage (S2). Each sample had at least 100 individual anthers. When the spikes (Figure 1j,q) were at Zadoks Stage 59, the first spike was visible, and the length of the spikes was more than 5 cm. The anthers were sampled for RNA-seq and described as the binucleate stage (S3). Each sample had at least 100 individual anthers. All the samples had 3 biological replicates. Every sample was directly dissected in the expeimental field, and frozen immediately in liquid nitrogen. Total RNAs were extracted majorly in our report [88] using TRIZOL reagent (TransGen Biotech, Beijing, China).

### 4.6. Transcriptome Sequencing and Data Analysis

The clean reads were mapped to the reference wheat genome version: IWGSC_RefSeq_v1.0. (https://urgi.versailles.inra.fr/download/iwgsc/IWGSC_RefSeq_Assemblies/v1.0/). Gene functional annotation was carried out as described by He et al. (2018) [88]. Gene expression levels were estimated by fragments per kilobase of transcript per million fragments mapped (FPKM) [89]. Differentially expressed genes (DEGs) between two sample pairs were analyzed using the DESeq R package [90]. The false discovery rate (FDR < 0.001) and fold change (FC ≥ 8) were set as the thresholds for DEGs. PCC and PCA were carried out to evaluate the indices of biological repetition correlation [91,92]. Log_2_FC ≥ 3 or ≤ −3 was set as the threshold parameter to screen for DEGs, and the expression values of the three replicates of each tissue were taken to perform gene co-expression analysis by *K*-means clustering algorithm [93]. If FPKM average value of a gene in the eighteen libraries was more than one, we defined it as a valid gene. The valid genes were used to carry out the statistical analyses of genes in different samples. All analyses were performed using software tools on BMKCloud (https://international.biocloud.net/zh/software/tools/). The bioproject accession of the transcriptome data in NCBI is PRJNA578055.

### 4.7. Weighted Gene Co-Expression Network Analysis (WGCNA)

Gene co-expression networks were constructed based on pairwise correlations between genes in their common expression trends across all sampled tissues using the WGCNA tool [94], which was a systemic analysis method aimed at understanding networks instead of individual genes. If the FPKM average value of a gene in the eighteen libraries was more than one, we defined it as a valid gene. R package WGCNA was conducted turning adjacency into topological overlap, which could measure the network connectivity of a gene defined as the sum of its adjacency with all other genes for network generation. Hierarchical clustering function was used to classify genes with similar expression profile into modules based on TOM dissimilarity with a minimum size of 30 for the gene dendrogram. The dissimilarity of module eigengenes was calculated to choose a cutline to merge some modules.

Hub genes that were highly interconnected with nodes in a module were considered functionally significant. kME was the value used to assess the effective connectivity between hub genes in each module. Each node in network had a degree, which was the number of edges associated with the node. DEGs in each module were chosen as key genes, and DEGs in the key modules were screened out for further analysis. The gene networks were visualized with Cytoscape.

### 4.8. qRT-PCR

The six samples of Shengnong 1 and NWMS1 at the S1, S2, and S3 stages were prepared for real-time PCR. The experimental samples were consistent with the samples of RNA-seq. All primers were designed using Primer Premier 5.0 software (www.premierbiosoft.com/primerdesign/index.html). The information of primers is listed in Appendix A. Reverse transcription was performed using TransScript^®^ All-in-One First-Strand cDNA Synthesis SuperMix for qPCR (TransGen Biotech, Beijing, China). Real time qRT-PCR was performed using TransStart^®^ Top Green Qpcr SuperMix (2×) (TransGen Biotech, Beijing, China), according to the manufacturer’s protocol on CFX ConnectTM Real-Time System (Bio-Rad, Hercules, CA, USA). The ingredients of the reaction system were strictly carried out according to the specified instructions. The reaction of quantitative RT-PCR and semi-quantitative RT-PCR were performed in 20 µL volumes [88]. The gene expression levels were calculated according to the 2^−∆∆*C*t^ method [95]. The wheat actin gene was used as an internal control.

## 5. Conclusions

Most pollen grains of mutant *NWMS1* collapse without starch grain accumulation. Apoptosis occurred for the cell nuclei of the remaining near-spheral pollen grains. The pollen abortion of *NWMS1* probably started at the early uninucleate microspore stage. The molecular regulation mechanisms of the abnormal anther and pollen developments in mutant *NWMS1* were investigated using RNA-seq and a series of experiments. A set of DEGs and a series of valuable metabolic pathways were identified, which were clearly associated with abnormal male development in *NWMS1*. They included protein processing in endoplasmic reticulum, starch and sucrose metabolism, plant hormone signal transduction, and 20 key genes as *SKP1B*, *BIP5*, *KCS11*, *ADH3*, *BGLU6*, and *TIFY10B*. The disturbed biological processes included starch and sucrose metabolism, lipid metabolism (fatty acid elongation, fatty acid degradation), phenylalanine metabolism, phenylpropanoid biosynthesis, and hormone signal transduction. The abnormally expressed typical genes included the homologs of *CYP703A2*, *CSA*, and *QRT3*. Disturbances of these biological processes and genes were documented as causative factors of plant male sterility. It was speculated that the disturbed starch and sucrose metabolism was the most important metabolism pathway causing male sterility in *NWMS1*. Based on the experimental data, a primary molecular regulation model of abnormal male development in *NWMS1* was established (Figure 10).

Although a number of genes involved in male development have been identified and functionally studied, the related genetic networks have not yet been fully established, and the elucidation of the underlying molecular mechanisms is still a big challenge. These results provide a basis for further exploration of the molecular mechanism of male sterility.

## Figures and Tables

**Figure 1 ijms-20-06252-f001:**
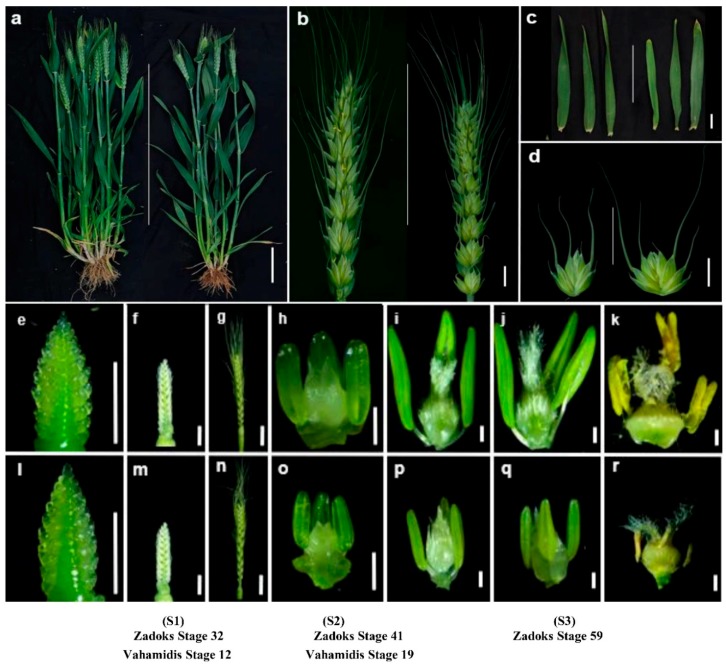
Morphology comparison of Shengnong 1 and mutant *NWMS1*. (**a**–**d**): Shengnong 1 (left) and mutant *NWMS1* (right). (**a**) individual plants; (**b**) the spikes in the anthesis stage; (**c**) the flag leaves in the heading stage; (**d**) middle spikelets on main stems in the anthesis stage; (**e**–**g**) young spikes of Shengnong 1; (**h**–**k**) stamens and pistils of Shengnong 1; (**l**–**n**) young spikes of mutant *NWMS1*; (**o**–**r**) stamens and pistils of mutant *NWMS1*. Scale bars: 10 cm (**a**); 1 cm (**b**–**d**); 0.5 cm (**e**–**r**).

**Figure 2 ijms-20-06252-f002:**
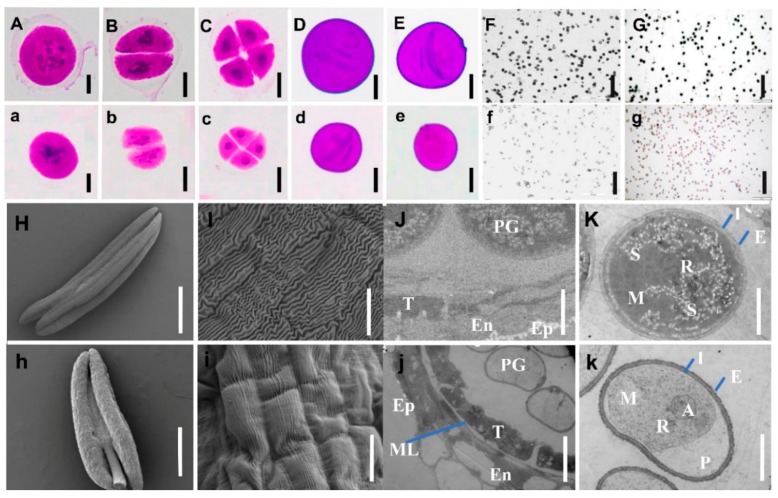
The anther and pollen microstructure and ultrastructure of Shengnong 1 (up) and mutant *NWMS1* (down). (**A**–**K**) Shengnong 1; (**a**–**k**) *NWMS1*. (**A**)/(**a**), meiotic interphase; (**B**)/(**b**), meiotic dyad; (**C**)/(**c**), meiotic tetrad; (**D**)/(**d**), (**E**)/(**e**), pollen grains; (**F**)/(**f**), pollen grains stained with 0.5% 2,3,5-triphenyltetrazolium chlorid (TTC) solution. (**G**)/(**g**), pollen grains stained with 1% KI-I_2_. (**H**)/(**h**), the complete anthers; (**I**)/(**i**), the epidermis of the anthers; (**J**)/(**j**), part anthers with pollen grains; (**K**)/(**k**), complete pollen grains. T, tapetum; Ep, epidermis; En, endothecium; ML, middle layer; PG, pollen grain; A, Autophagosome; E, exine; I, intine; M, mitochondrion; *p*, plasmolysis; R, rough endoplasmic reticulum; S, starch grain. Scale bars: 1 mm in (**H**)/(**h**); 30 μm in (**I**)**/**(**i**); 10 μm in (**J**–**K)**/(**j**–**k**); 20 μm in (**A**–**E**)/(**a**–**e**); 200 μm in (**F**–**G**)/(**f**–**g**).

**Figure 3 ijms-20-06252-f003:**
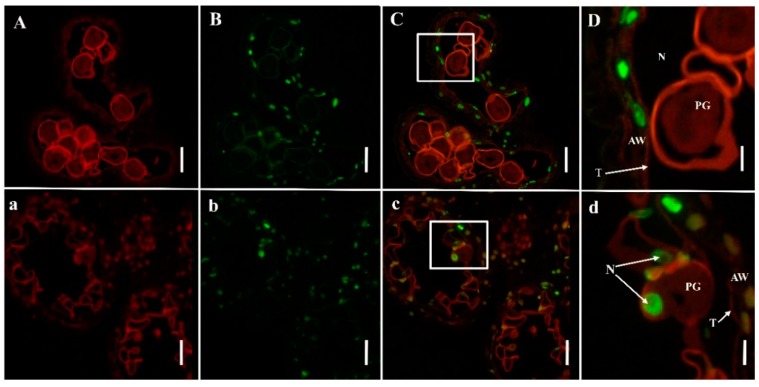
TUNEL assays to detect anther programmed cell death (PCD) in Shengnong 1 (up) and mutant *NWMS1* (down) when the microspores were at the binucleate stage. (**A**)/(**a**), anthers stained with DAPI; (**B**)/(**b**), anthers stained with TUNEL kit; (**C**)/(**c**), overlapped pictures of the anthers stained with DAPI and TUNEL kit; (**D**)/(**d**), enlarged pictures of C and c. N, nucleus; T, tapetum; PG, pollen grain; AW, anther wall, The nucleus stained with DAPI was red, positive apoptotic cell nuclei stained with TUNEL kit was deep green. Scale bars: 30 μm in (**A**–**C**)/(**a**–**c**); 10 μm in (**D**)/(**d**).

**Figure 4 ijms-20-06252-f004:**
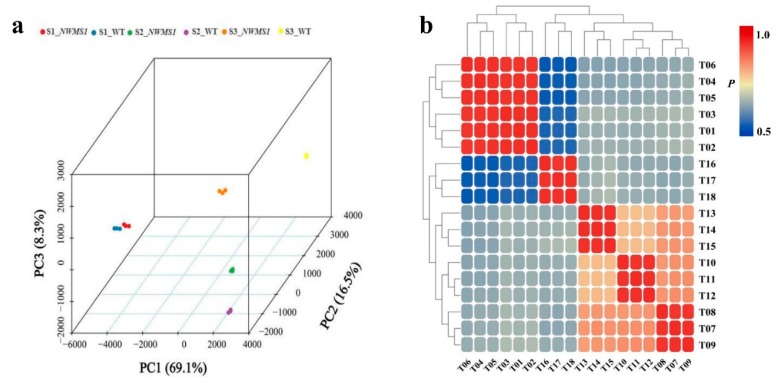
Correlations among the samples. (**a**) the PCA map of the samples; (**b**) PCC map of the samples.

**Figure 5 ijms-20-06252-f005:**
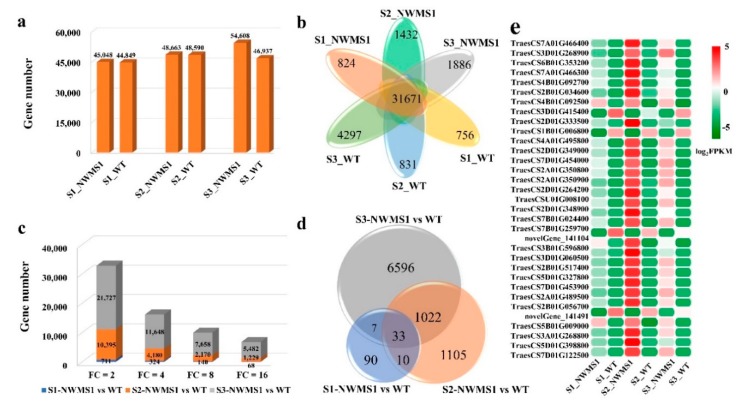
The differentially expressed genes’ profiles. (**a**) a histogram of the valid gene numbers in the six samples; (**b**) a Venn diagram of the valid gene numbers in six samples; (**c**) the DEG numbers with different fold changes. (**d**) statistics of DEGs among samples. (**e**) a heat map of the 33 DEGs in six samples.

**Figure 6 ijms-20-06252-f006:**
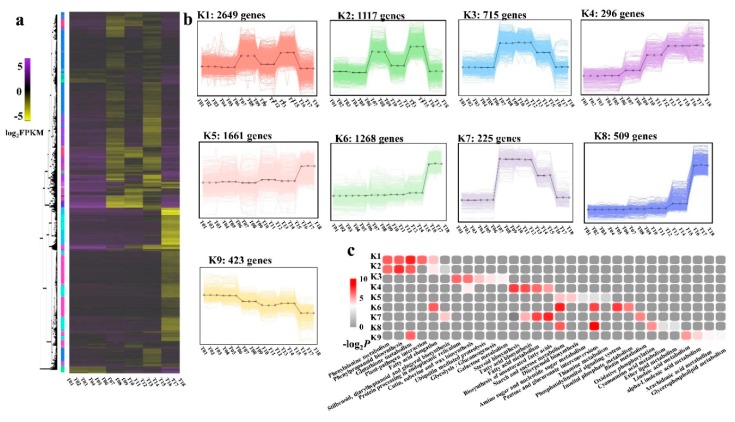
The hierarchical clusters of the DEGs identified among the eighteen libraries. (**a**) Heatmap of the DEGs among the eighteen libraries. The gene expression values were presented as log_2_-transformed normalized FPKM values. Nine clusters were shown. (**b**) K1-K9, the nine clusters of the DEGs. (**c**) the top five enrichment pathways of the DEGs in different clusters, referring to the KEGG database.

**Figure 7 ijms-20-06252-f007:**
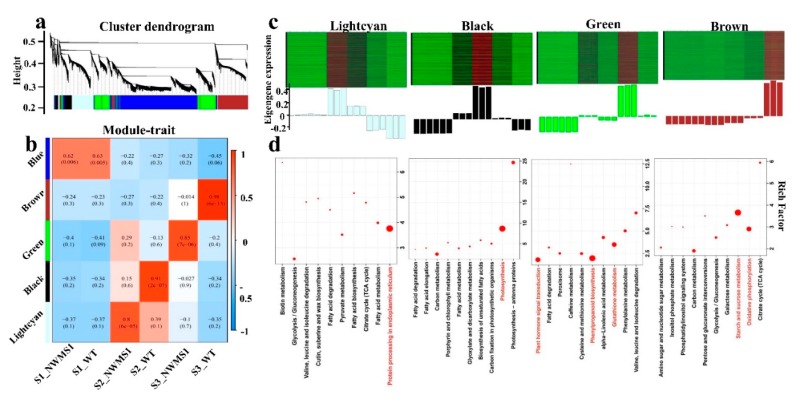
The co-expression modules of the valid genes analyzed via weighted gene co-expression network analysis (WGCNA). (**a**) the co-expression profiles of the valid expressed genes. (**b**) the correlation coefficients between the modules and the samples. The numbers in each box represented the correlation coefficient (up) and standard error (down); (**c**) the eigengene expression levels in different modules. (**d**) the enriched KEGG pathways of the eigengenes in each module.

**Figure 8 ijms-20-06252-f008:**
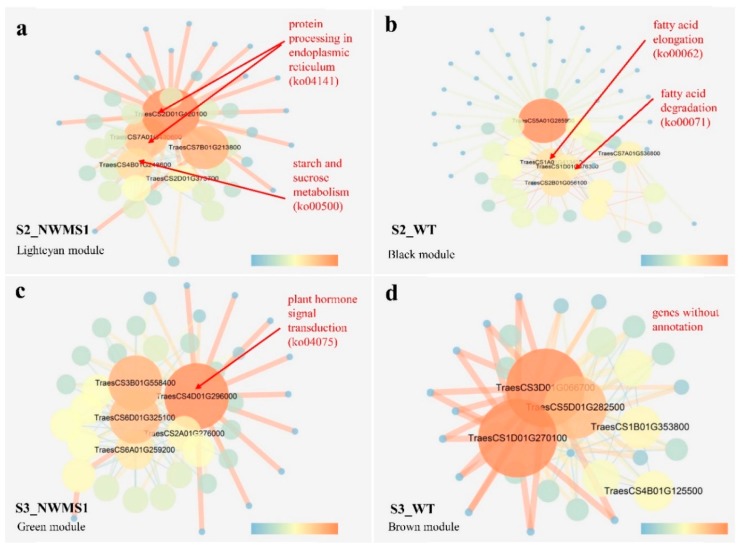
The co-expression networks of the DEGs between *NWMS1* and Shengnong 1 in the four important modules established via WGCNA. (**a**) the network in the brown module; (**b**) the network in the green module; (**c**) the network in the black module; (**d**) the network in the light cyan module. The colour gradation from dark to light represents the size of the nodes and the number of edges.

**Figure 9 ijms-20-06252-f009:**
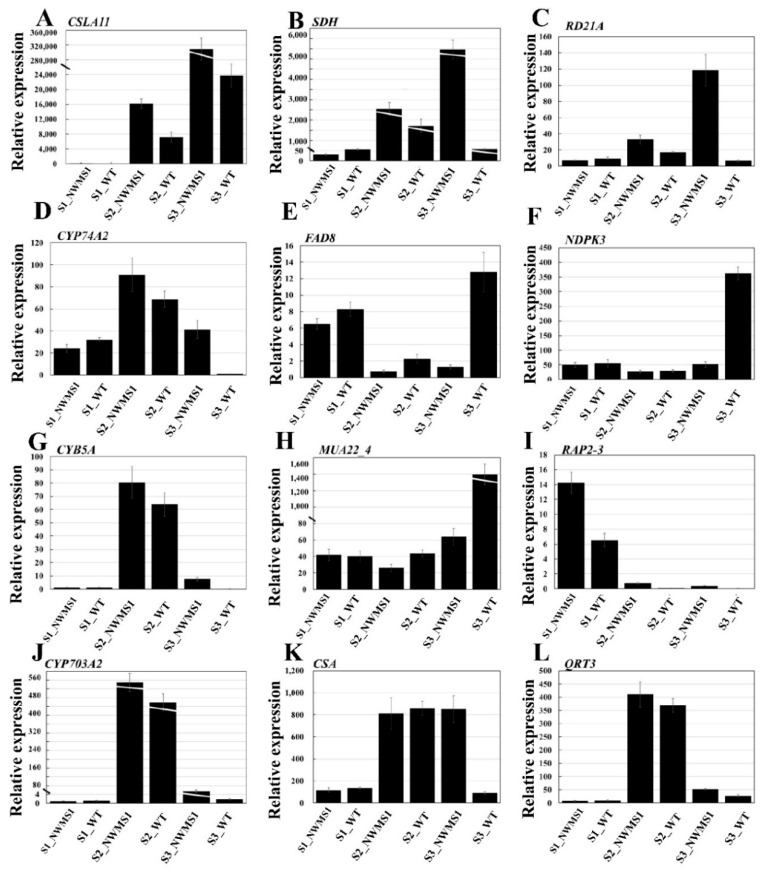
Spatiotemporal expression profiles of the twelve genes. **(****A**) TraesCS3B01G558400 (probable mannan synthase 11); (**B**) TraesCS6D01G325100 (saccharopine dehydrogenase); (**C**) TraesCS1D01G327000 (cysteine proteinase RD21a); (**D**) TraesCS4A01G061900 (allene oxide synthase 2); (**E**) TraesCS4A01G182700 (temperature-sensitive omega-3 fatty acid desaturase, chloroplastic); (**F**) TraesCS1B01G478800 (nucleoside diphosphate kinase III, chloroplastic/mitochondrial); (**G**) TraesCS7A01G437700 (cytochrome b5); (**H**) TraesCS2B01G335500 (mitochondrial phosphate carrier protein 3, mitochondrial); (**I**) TraesCS1D01G230900 (ethylene-responsive transcription factor RAP2-3); (**J**) TraesCS7A01G309300 (cytochrome P450 703A2); (**K**) TraesCS2D01G407700 (probable phospholipid hydroperoxide glutathione peroxidase); (**L**) TraesCS2A01G423900 (polygalacturonase QRT3).The *actin* gene was used as internal control. The functional annotation and other details of A–L were listed in Appendix A. All qRT-PCR reactions were replicated thrice.

**Figure 10 ijms-20-06252-f010:**
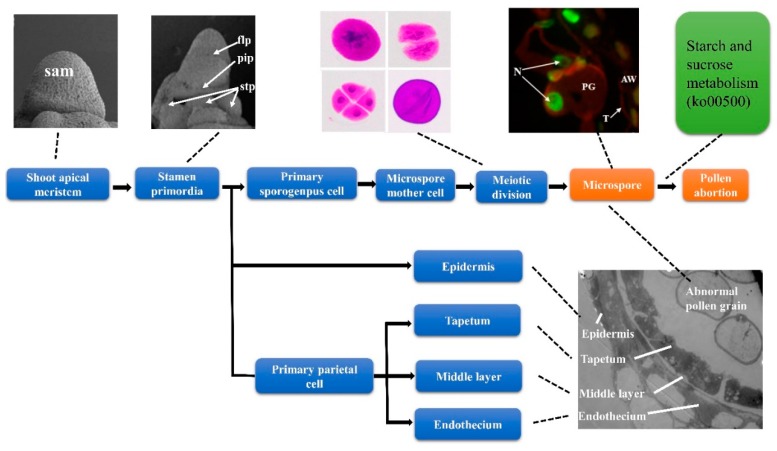
Male development of *NWMS1* and the molecular factors causing pollen abortion. sam, shoot apical meristem; flp, floret primordium; stp, stamen primordium; pip, pistil primordium; N, nucleus; T, tapetum; PG, pollen grain; AW, anther wall.

**Table 1 ijms-20-06252-t001:** Comparison of agronomic traits between Shengnong 1 and mutant *NWMS1*, 2017–2018.

Traits	Shengnong 1	*NWMS1*
Plant height (cm)	70.59 ± 1.54	68.19 ± 3.45 *
Spike length (cm)	10.6 ± 0.39	8.73 ± 0.51 *
Flag leaf length (cm)	20.02 ± 3.37	18.03 ± 2.03 *
Flag leaf width (cm)	2.3 ± 0.15	1.88 ± 0.19 *
Internode length under spike (cm)	31.83 ± 2.39	29.24 ± 5.21 *
Effective tiller number	13.4 ± 2.55	10.2 ± 2.53 *
Internode number of the main stem	5.00 ± 0.00	5.00 ± 0.00
Spikelet number on the main stem	20.4 ± 1.35	19.4 ± 1.84
Spike type	Rectangle	Rectangle
Awn color	White	White
Heading stage (Month-Day)	04-10	04-17 **
Anthesis stage (Month-Day)	04-23	05-01 **
Maturity stage (Month-Day)	05-22	05-31 **

* significant at *p* = 0.05; ** significant at *p* = 0.01.

**Table 2 ijms-20-06252-t002:** The major enrichment terms of the DEGs at different stages, with reference to the GO database.

Sample Pair	Class	Annotation	GO ID	Corrected *p*-Value
S1-*NWMS1* vs. WT	biological process	response to cadmium ion	GO:0046686	1.9 × 10^−23^
	cellular component	cytoplasmic membrane-bounded vesicle	GO:0016023	1.0 × 10^−30^
	molecular function	electron carrier activity	GO:0009055	3.9 × 10^−19^
S2-*NWMS1* vs. WT	biological process	pentose-phosphate shunt	GO:0006098	1.0 × 10^−30^
	cellular component	cytoplasmic vesicle	GO:0016023	1.0 × 10^−30^
	molecular function	heme binding	GO:0020037	2.2 × 10^−22^
S3-*NWMS1* vs. WT	biological process	oxidation-reduction process	GO:0055114	4.8 × 10^−25^
	cellular component	cytoplasmic membrane-bounded vesicle	GO:0016023	1.0 × 10^−30^
	molecular function	heme binding	GO:0020037	1.6 × 10^−29^

**Table 3 ijms-20-06252-t003:** Major pathways of the DEGs at different stages, with reference to the KEGG database.

Sample Pair	Pathway	Ko ID	Corrected *p*-Value
S1-*NWMS1* vs. WT	amino sugar and nucleotide sugar metabolism	ko00520	0.031039284
S2-*NWMS1* vs. WT	phenylalanine metabolism	ko00360	0
	phenylpropanoid biosynthesis	ko00940	0
	glutathione metabolism	ko00480	1.50 × 10^−10^
S3-*NWMS1* vs. WT	phenylalanine metabolism	ko00360	0
	glutathione metabolism	ko00480	0
	starch and sucrose metabolism	ko00500	0

**Table 4 ijms-20-06252-t004:** The information of key genes involved in abnormal anther development obtained via WGCNA.

Gene ID	Modules	Clusters	S1_*NWMS1*	S1_WT	S2_*NWMS1*	S2_WT	S3_*NWMS1*	S3_WT	Gene Name
TraesCS2D01G420100	light cyan	K3	0.00	0.00	21.36	14.53	5.23	0.50	*SKP1B*
TraesCS7B01G213800	light cyan	K3	0.00	0.02	3.35	2.62	0.81	0.02	*Os03g0802700*
TraesCS7A01G430600	light cyan	K3	0.01	0.00	21.98	16.60	5.20	0.09	*BIP5*
TraesCS4B01G248600	light cyan	K3	2.07	2.56	43.93	34.65	12.54	0.68	*BGLU6*
TraesCS2D01G373700	light cyan	K3	0.00	0.00	41.16	23.75	4.36	0.42	*F775_16360*
TraesCS5A01G285900	black	K3	0.00	0.00	2.95	5.69	2.04	0.03	*PAP27*
TraesCS1A01G413400	black	K3	0.00	0.00	9.60	18.89	5.89	0.49	*KCS11*
TraesCS2B01G056100	black	K3	0.00	0.00	23.47	43.09	11.88	0.05	*SWEET6B*
TraesCS1D01G376300	black	K3	0.09	0.10	8.40	17.96	6.75	0.25	*ADH3*
TraesCS7A01G536800	black	K3	0.01	0.00	8.33	32.95	6.53	0.12	*At5g03610*
TraesCS4D01G296000	green	K2	0.41	0.04	41.70	16.73	65.78	5.44	*TIFY10B*
TraesCS3B01G558400	green	K2	0.60	0.52	8.20	5.28	15.95	1.75	*CSLA11*
TraesCS6D01G325100	green	K2	0.54	0.89	44.31	29.72	94.55	9.71	*SDH*
TraesCS6A01G259200	green	K2	0.21	0.02	18.59	7.12	31.49	2.80	*ATL41*
TraesCS2A01G276000	green	K2	0.10	0.09	4.03	1.87	7.64	0.69	*EXO70B1*
TraesCS1D01G270100	brown	K4	0.06	0.00	0.25	8.62	112.57	445.69	*GASA2*
TraesCS3D01G066700	brown	K4	0.00	0.00	1.20	10.69	126.30	507.99	*F775_31707*
TraesCS5D01G282500	brown	K4	0.00	0.01	0.09	1.87	34.21	146.67	*SAG12*
TraesCS1B01G353800	brown	K4	0.00	0.01	0.04	0.74	21.35	94.06	*STP8*
TraesCS4B01G125500	brown	K6	0.00	0.00	0.04	0.56	4.34	23.94	*At1g26850*

The modules represented the different color modules in the WGCNA results; clusters represented the different clusters in the *K*-means analysis results.

**Table 5 ijms-20-06252-t005:** The expression profiles of key anther development-related genes between mutant *NWMS1* and Shengnong 1.

Gene ID	S1_*NWMS1*	S1_WT	S2_*NWMS1*	S2_WT	S3_*NWMS1*	S3_WT	Gene Name	Reported Function
novelGene_53829	0.00	0.00	0.22	0.03	0.01	0.02	*PAIR1*	3
novelGene_56612	0.20	0.23	0.09	0.20	0.22	0.16	*PAIR2*	3
TraesCS7A01G316600	1.40	1.26	0.79	0.56	0.96	1.11	*PSS1*	3, 5
TraesCS2A01G226000	8.31	8.92	2.03	2.59	1.98	1.69	*BUB1*	2, 3
TraesCS6A01G364300	8.81	9.02	1.34	1.02	1.60	1.39	*SGO1*	2, 3
**TraesCS7A01G309300**	**0.48**	**0.47**	**260.82**	**220.0**	**26.93**	**0.83**	***CYP703A2***	**4, 5, 6**
novelGene_128161	0.54	0.68	3.13	1.28	0.56	0.26	*MYB80*	4, 6
novelGene_128276	0.00	0.07	0.08	0.00	0.06	0.09	*DEX1*	4, 5
TraesCS1B01G247200	4.19	5.16	9.20	8.53	9.42	7.83	*DAD1*	5, 6
TraesCS7A01G305700	2.57	2.33	0.36	0.59	0.19	0.24	*TDR*	4
TraesCS7A01G458700	0.00	0.00	0.00	0.03	0.17	25.50	*GAMYB*	5, 6
TraesCS5A01G481600	0.00	0.00	0.00	0.00	0.00	0.00	*BAM1*	1
TraesCS1A01G444500	3.78	3.90	0.23	0.11	0.18	0.12	*BAM2*	1
**TraesCS2D01G407700**	**28.12**	**32.21**	**183.32**	**193.7**	**192.5**	**20.81**	***CSA***	**6, 7**
novelGene_21887	0.02	0.00	0.01	0.00	0.00	0.20	*TDL1B*	1, 3
novelGene_139463	2.67	2.34	3.78	2.67	2.63	2.35	*MSP1*	1, 3
**TraesCS2A01G423900**	**1.95**	**2.32**	**91.01**	**82.45**	**11.51**	**5.97**	***QRT3***	**6**

1, early anther development; 2, mitosis; 3, meiosis; 4, tapetum; 5, pollen wall development; 6, microspore differentiation; 7, intercellular signal transductions.

**Table 6 ijms-20-06252-t006:** Expression changes of some DEGs encoding key enzymes involved in starch and sucrose metabolism at three stages.

Enzyme D	Enzyme Code *	S1-Log_2_ (*NWMS1/*WT) **	S2-Log_2_ (*NWMS1/*WT)	S3-Log_2_ (*NWMS1/WT*)
glycogen phosphorylase	2.4.1.1	−	−1.2	−5.64
sucrose synthase	2.4.1.13	−	−2	5.07 & −2.0
sucrose-phosphate synthase	2.4.1.14	−	−2.5	−1.24
trehalose 6-phosphate synthase	2.4.1.15	−	1.2 & −1.5	4.48 & −2.86
1,4-alpha-glucan branching enzyme	2.4.1.18	−	−	1.43 & −3.96
starch synthase	2.4.1.21	−	−1.2	1.24 & −1.79
4-alpha-glucanotransferase	2.4.1.25	−	−1.23	−2.72
Hexokinase	2.7.1.1	−	3.2 & −1.0	5.14 & −3.74
Fructokinase	2.7.1.4	−	1.2 & −1.5	3.62 & −2.58
glucose-1-phosphate adenylyltransferase	2.7.7.27	−	−1.2	−3.56
UTP-glucose-1-phosphate uridylyltransferase	2.7.7.9	−	−	−8.72
trehalose 6-phosphate phosphatase	3.1.3.12	1.67 & −10.25	3.0 & −1.5	3.16 & −1.68
alpha-amylase	3.2.1.1	−	−2.78	9.25 & −1.30
beta-amylase	3.2.1.2	−	1	2.10 & −3.89
alpha-glucosidase	3.2.1.20	−	1.8	1.99 & −2.03
beta-glucosidase	3.2.1.21	1.47	−	2.54 & −2.65
eta-fructofuranosidase	3.2.1.26	−1.01	1.1 & −2.1	1.67 & −3.28
alpha, alpha-trehalase	3.2.1.28	−	−	2.46
glucan endo-1,3-beta-glucosidase 1/2/3	3.2.1.39	−1.88	1.2 & −1.7	1.92 & −1.56
Endoglucanase	3.2.1.4	−	2	−6.85
ADP-sugar diphosphatase	3.6.1.21	−	−	−2.75
Phosphoglucomutase	5.4.2.2	−	−	−1.71

* Enzyme code is a unique code for each enzyme in KEGG database; the codes can be found at https://www.kegg.jp. ** The logarithm value of the fold change of gene expression value of the DEGs between *NWMS1* and WT. ‘−’ represents no significant expression difference at the corresponding stage. Gene expression levels were estimated by fragments per kilobase of transcript per million fragments mapped (FPKM).

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
