# Peer review of "The Major Factors Causing the Microspore Abortion of Genic Male Sterile Mutant NWMS1 in Wheat (Triticum aestivum L.)"

_ijms, 2019, doi:10.3390/ijms20246252_

Round 1
Reviewer 1 Report
The MS presented very useful insights on male sterility in wheat. The methods are sound and results are presented well.
The only limitation is that English language should be improved.
Author Response
Point 1: The only limitation is that English language should be improved. 

Response 1: We have reviewed the manuscript carefully again and checked through the whole manuscript and corrected a few grammar mistakes. At the same time, we invite professionals for language polish, too. Thank you very much for your valuable comments.

Reviewer 2 Report
This article described the characteristics at morphological and molecular levels of the genic male sterile mutant NWMS1. The authors clearly define the moment in which the abortion process started (i.e. at the early uninucleate microspore stage). RNA-seq approach was used to describe which metabolic pathways changed in the mutant, so to obtain new information on the molecular bases of abortion that could operate in wheat.
The main conclusion of the authors was that the cause of abortion is strictly linked to deficiency in starch and sucrose metabolism (i.e. a deficiency to synthetize starch). Considering that this is the main conclusion of this article, the authors must show better the molecular data regarding sugar/starch pathways. Considering the molecular approach used as well as the number of the genes detected, it is realistic that the expression of many genes codifying for sugar transporters and/or enzymes involved in the starch biosynthesis have been detected and that they showed a different expression in the two genotypes (Shengnong 1 versus NWMS1). In other words, the authors should introduce a table reporting these results (to report the expression of the proteins involved in starch biosynthesis, such as hexose transporters, PM H+-ATPase, soluble invertase 2, Suc synthase, Suc 6-phosphate phosphohydrolase, hexokinase, phosphoglucomutase, ADP-Glc pyrophosphorylase, granule-bound starch synthase, etc…). Certainly, the authors must consider the enzymes that are known to be present/operating in wheat. In this version of article, in fact, the reader does not clearly see if these pathways are affected in the mutant.
In this view, the authors could consider the work of Datta and co-workers (www.plantphysiol.org/cgi/doi/10.1104/pp.006908), which study the same problematic in maize and that formulated a same conclusion.
Author Response
Point 1: The authors must show better the molecular data regarding sugar/starch pathways.
Response 1:
In order to show the results of metabolic pathways to the readers clearly, we have added Table 6 to this revised version. Table 6 listed the significant DEGs encoding key enzymes involved in starch and sucrose metabolism pathway. Meanwhile, we have added Figure S3, Figure S4 and Figure S5 as supplementary information for Table 6. Besides, related discussion has been added in section ‘4.4. Starch and sucrose metabolism is the major factor causing male sterility in NWMS1’. Thank you very much for your valuable comments.
